# Spatial disparities in the mortality burden of the covid-19 pandemic across 569 European regions (2020-2021)

Florian Bonnet [1] ✉, Pavel Grigoriev [2], Markus Sauerberg [2], Ina Alliger[2], Michael Mühlichen [2] & Carlo-Giovanni Camarda[1]

Since its emergence in December 2019, the COVID-19 pandemic has resulted in a significant increase in deaths worldwide. This article presents a detailed analysis of the mortality burden of the COVID-19 pandemic across 569 regions in 25 European countries. We produce age and sex-specific excess mortality and present our results using Age-Standardised Years of Life Lost in 2020 and 2021, as well as the cumulative impact over the two pandemic years. Employing a forecasting approach based on CP-splines that considers regional diversity and provides confidence intervals, we find notable losses in 362 regions in 2020 (440 regions in 2021). Conversely, only seven regions experienced gains in 2020 (four regions in 2021). We also estimate that eight regions suffered losses exceeding 20 years of life per 1000 population in 2020, whereas this number increased to 75 regions in 2021. The contiguity of the regions investigated in our study also reveals the changing geographical patterns of the pandemic. While the highest excess mortality values were concentrated in the early COVID-19 outbreak areas during the initial pandemic year, a clear East-West gradient appeared in 2021, with regions of Slovakia, Hungary, and Latvia experiencing the highest losses. This research underscores the importance of regional analyses for a nuanced comprehension of the pandemic's impact.

In 2023, the number of deaths due to COVID-19 was much lower than in the years 2020 to 2022. The WHO, therefore, declared the end of the global health emergency on 6 May 2023. It is now time to evaluate the overall burden of the pandemic, particularly in the years 2020 and 2021, when it was at its peak.

To do this, scholars have first used reports of case fatalities published by national surveillance authorities[1,2] but are now mainly calculating excess mortality, defined as "the difference between the number of deaths (from any cause) that occur during the pandemic and the number of deaths that would have occurred in the absence of the pandemic"[3]. This is considered to be the gold standard for estimating the overall impact of COVID-19[4,5], and especially more reliable

than deaths coming from epidemiological surveillance data, due to different definitions of data among countries, time-varying collection methods, reporting delays, and diverse comprehensiveness by place of death[6,7].

Many studies have attempted to quantify the impact of the pandemic using this approach. However, most of them have done so at national level[8–17]. A few other studies have attempted to quantify the impact of the pandemic at a finer geographical scale, but for one country at a time[18–27]. However, comparing these regional patterns is problematic because these studies take different approaches to compute the mortality levels that would have occurred without the pandemic. Specifically, they either use pre-pandemic levels or employ

[1]French Institute for Demographic Studies (INED), Aubervilliers, France. [2]Federal Institute for Population Research (BiB), Wiesbaden, Germany. ✉e-mail: florian.bonnet@ined.fr

forecasting techniques. Moreover, these papers rely on different indicators to assess excess mortality, e.g. life expectancy or death toll. It is therefore impossible to use these results to compare the impact of the pandemic between regions in one country and those in another. To our knowledge, only two peer-reviewed studies allow for a simultaneous comparison of regional excess mortality in several European countries for 2020[28,29]. Another peer-reviewed study covers 200 NUTS 2 European regions for the years 2020 to 2022 but does not estimate excess mortality in regions of Germany, the UK, Ireland or Sweden[30]. The ONS has also published on its website a report[31] on weekly excess mortality between the end of 2019 and mid-2022 by region in Europe. Finally, most of these studies quantified the impact of the COVID-19 pandemic in 2020 only, while the virus was still virulent in 2021.

However, it is important to produce these estimates at a fine geographical level, because the pandemic affected in different ways the regions of the same country: for example, the North of Italy was severely affected by the pandemic in 2020 while the South was mostly spared[18]. These differences can be explained in particular by the locations where the virus first arrived in Europe, and by the travel restrictions that were enforced to prevent the spread of the virus over space. It is quite likely that spatial differences would still be visible in 2021, due in particular to the spread of the virus strains or to the differences in cultural resistance to the vaccination campaign launched that year[32].

From a methodological point of view, assessing the full impact of the pandemic in 2020 and 2021 is a challenge. The death toll, widely used in studies, fails to consider differences between population's age structures. For this reason, scholars prefer looking at age-specific mortality rates instead, as they can be aggregated into a summary measure such as the age-standardised death rate or period life expectancy. However, for assessing the total burden of the pandemic while differentiating 2020 from 2021, life expectancy and age-standardised death rates are not convenient candidates as they cannot be added up over time.

Our paper aims to fill these research gaps by presenting Age-Standardised Years of Life Lost (ASYLL) in 2020 and 2021 for 569 regions of comparable size from 25 countries in Central and Western Europe. To obtain the mortality levels that would have been observed in 2020 and 2021 in the absence of the pandemic, we took a robust forecasting approach that accounts for regional diversity and delivers confidence intervals surrounding our excess mortality measures. Calculating these confidence intervals is crucial for robust and reliable data interpretation: fine-grained analyses often involve small populations, making data susceptible to increased variability. At the end, we reveal to what extent the European regions suffered from the COVID-19 pandemic in 2020 and 2021, and how the geographic patterns of excess mortality due to the pandemic have changed between these two years.

## Results

We begin with the assessment of the spatial disparities in ASYLL observed across 569 spatial units in 25 European countries during the first (2020) and the second (2021) year of the COVID-19 pandemic among males (Fig. 1) and females (Fig. 2). The regions highlighted in light blue colour are those that experienced gains in years of life compared to the expected values, whereas the remaining colours of the legend imply the age-adjusted years of life losses per 1000 population. Supplementary Fig. 1 in supplementary appendix A reveals ASYLL variations between 2020 and 2021.

During the first pandemic year, high ASYLL were mostly located in northern Italy, southern Switzerland, central Spain, and Poland. The highest ASYLL (above 30 years of life lost per 1000 population) were observed in those places where the outbreak of the COVID-19 was first reported in Europe (Italy and Spain). By contrast, the majority of

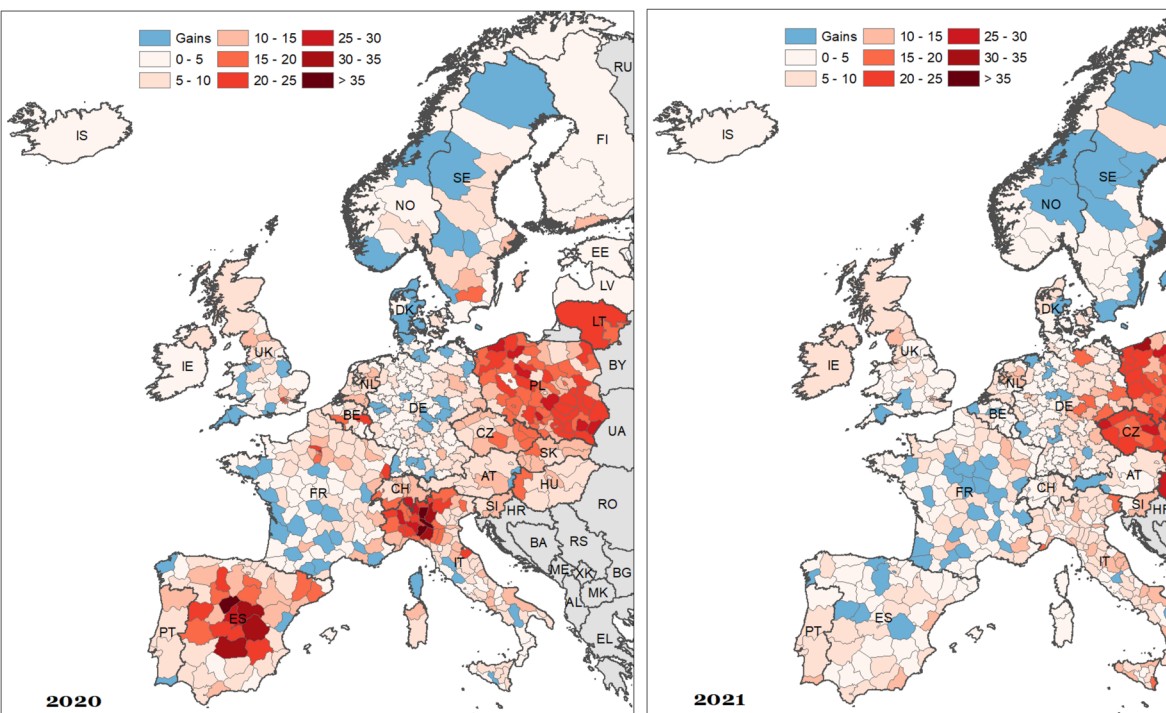

**Fig. 1 | Spatial distribution of age-standardised years of life lost (ASYLL) across 25 European countries in 2020 and 2021, males (per 1000 population).** Notes: ASYLL quantifies excess mortality in terms of life years lost. It calculates the potential additional mortality in a given period, associates this age-specific excess mortality with the number of years the population would have lived, and finally sums these values up standardising them with respect to a reference age structure.

Thus, ASYLL is unaffected by the population size and age structure of the underlying population. Age-specific excess mortality is defined as the difference between forecasted mortality rates based on the pre-pandemic mortality trend and the mortality rates observed in the pandemic years 2020 and 2021. Source data are provided as a Source Data file.

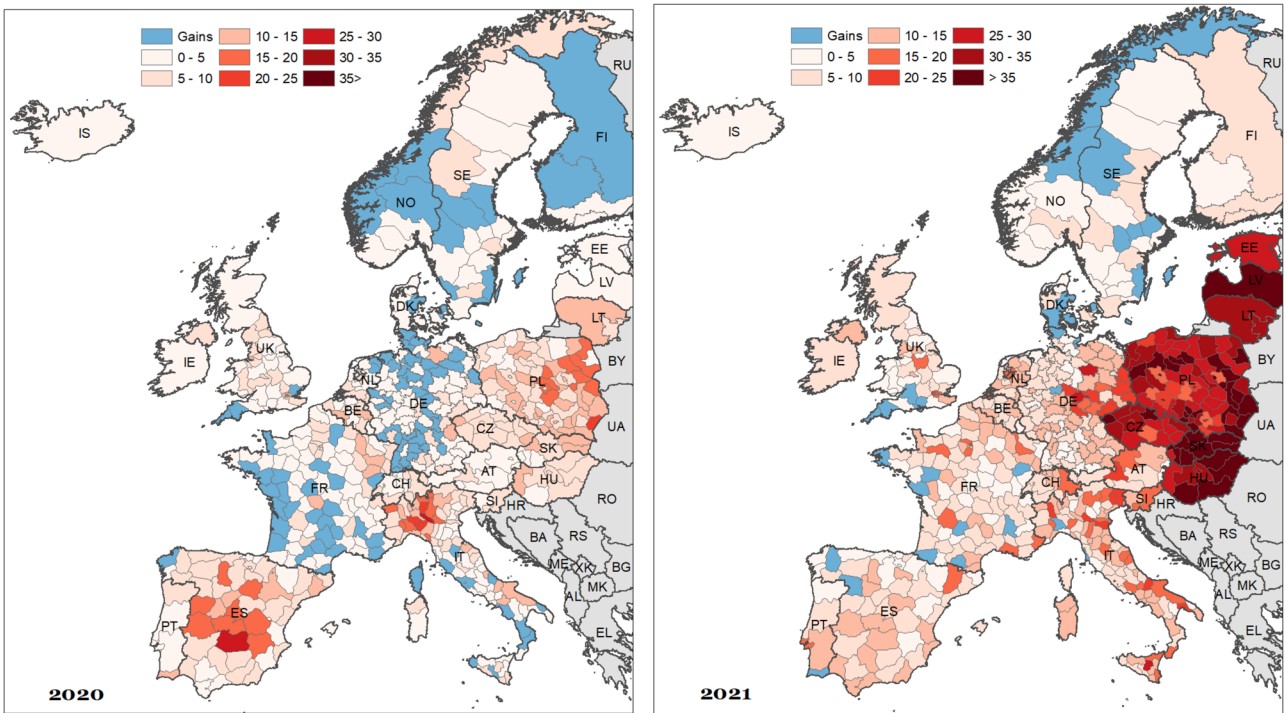

**Fig. 2 | Spatial distribution of age-standardised years of life lost (ASYLL) across 25 European countries in 2020 and 2021, females (per 1000 population).** Notes: As for Fig. 1. Source data are provided as a Source Data file.

French and German regions, the south of the UK, as well as Finland, Iceland, Northern Ireland, Estonia, Latvia, and Hungary experienced modest losses. Negative values of ASYLL (i.e. reduction of mortality compared to the baseline), and, thus, gains in years of life were concentrated in western and southwestern France as well as Denmark, Norway, and Sweden.

The spatial patterns of excess mortality in Europe changed drastically during the second pandemic year. In 2021, the highest losses were observed exclusively in the Eastern European countries, and particularly among men. As far as male mortality is concerned, the highest losses were observed in Slovakia, Hungary, and Latvia (more than 35 years of life lost per 1000 population). Unlike the spatial patterns of excess mortality in 2020, those observed in 2021 followed closely the known East-West mortality gradient[33]. The East-West differences are particularly pronounced in female excess mortality. Almost all regions located in Western Europe experienced rather moderate losses or even gains in years of life. It is interesting to note that the German-Polish and German-Czech borders do not clearly demarcate the zones of high and modest excess mortality: they present intermediate losses between neighbouring Czechia and Poland and the western part of Germany.

Finally, and using values for both sexes combined, we found evidence of gains in only seven regions in 2020, and four regions in 2021. On the contrary, 362 regions experienced notable losses during the first year of the pandemic, and 440 regions in the second year. Importantly, only eight regions suffered from ASYLL higher than 20 per 1000 population in 2020, while 75 regions suffered such a loss in 2021.

Figure 3 depicts the combined effects of the two pandemic years, which were particularly pronounced among men. Most of the regions with the highest male excess mortality were predominantly located in Poland, Slovakia, and Lithuania. The ASYLL values in the other countries of Eastern Europe were also high but comparable to those observed in northern Italy and central Spain. Among the Baltic States, Estonia experienced the lowest male excess mortality over the two pandemic years. This excess mortality was comparable in magnitude

to the one observed in the regions of eastern Germany, northern Austria, Slovenia, central and northern Italy, Switzerland, the Netherlands as well as to several French regions close to the Belgian and German borders. Among 25 countries considered here, the majority of regions located in Scandinavian countries, western and northern Germany, and France experienced either relatively small losses or even gains during 2020–2021. The favourable trend in these regions is more pronounced among women.

Using values for both sexes combined, we estimate that only two regions experienced a significant gain during the two years of our study, while 458 regions exhibit a notable loss; among them, 136 suffered from ASYLL higher than 20 per 1000 population. Our results also show that more than twice as many regions (151) experienced high excess mortality among men as among women (73) during the two years of the pandemic.

The analysis of the combined effects of the two pandemic years indicates that excess mortality was generally higher in the countries of Central and Eastern Europe (CEE) with lower life expectancy before the pandemic. However, we also notice that in 2020, the highest excess mortality was observed in European regions with the highest pre-pandemic life expectancy at birth, such as central Spain and northern Italy. In what follows, we examine the ecological associations between the magnitude of excess mortality due to the COVID-19 pandemic and the initial mortality level across 569 spatial units. Our hypothesis is that regions with higher initial mortality during the pre-pandemic years (2015-2019) experienced higher excess mortality in 2020 and 2021. In this context, baseline life expectancy serves as a proxy of 'initial conditions' reflecting different aspects related to population health such as the quality and accessibility of health care, the level of socioeconomic development, the prevalence of risk factors and disease burden, environmental conditions, etc. We stratify the analysis by year (2020 and 2021) and broader geographical regions (CEE and West) to account for the substantial differences between the two pandemic years as well as the differences in mortality levels between the CEE countries and remaining Europe (Fig. 4).

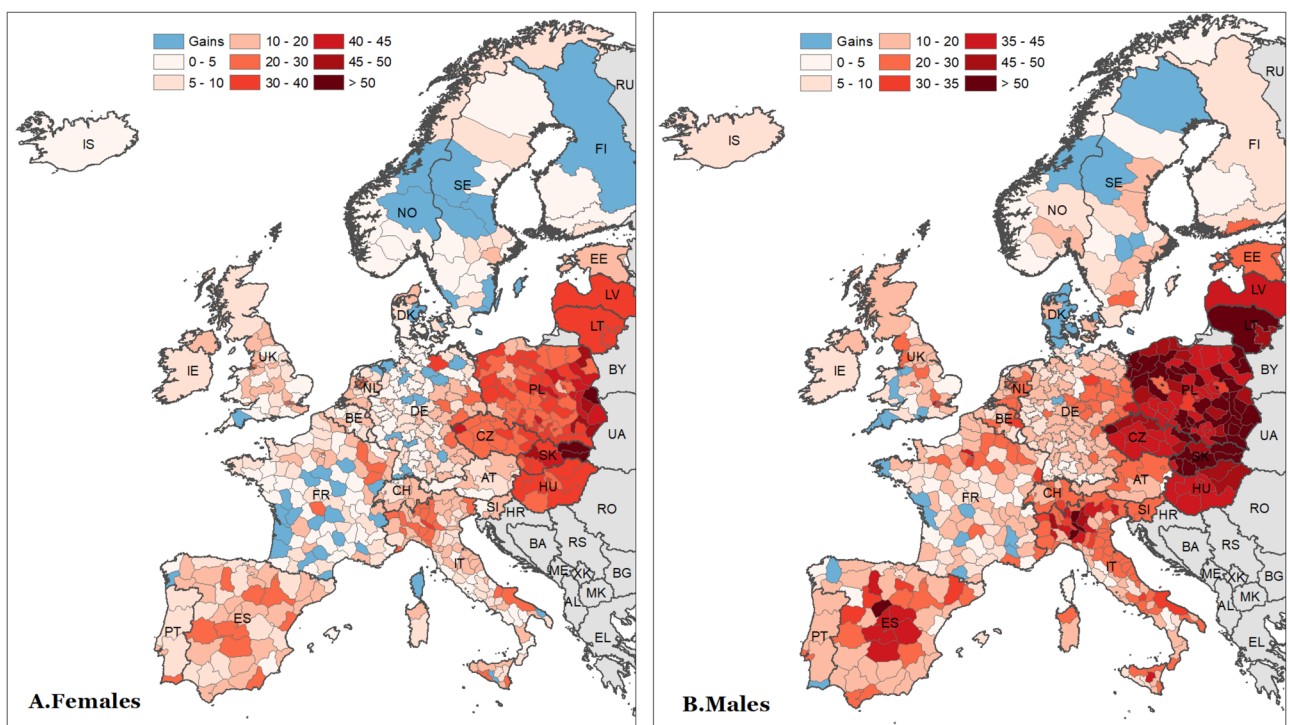

**Fig. 3 | Spatial distribution of age-standardised years of life lost (ASYLL) across 25 European countries during 2020–2021 (per 1000 population).** Notes: As for Fig. 1. Source data are provided as a Source Data file.

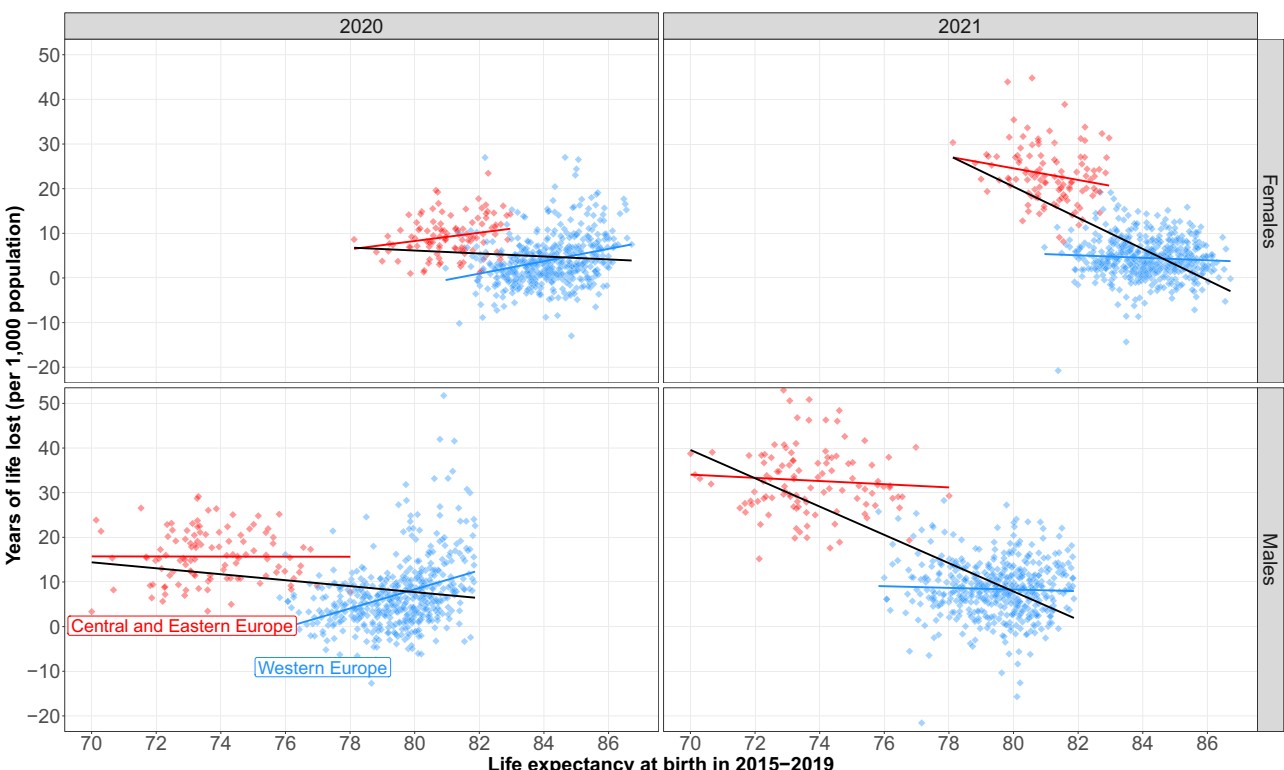

**Fig. 4 | Age-standardised years of life lost (ASYLL) against life expectancy at birth in 2015–2019 across 25 European countries.** Notes: The lines show the linear relationship between ASYLL and life expectancy at birth in the period 2015–2019. Red-coloured points correspond to regions located in Central and Eastern Europe, while blue-coloured points refer to regions located in Western Europe. The black line reflects the relationship for all points in red or blue. ASYLL quantifies the level of excess mortality in terms of life years lost (see the notes of Fig. 1 for more details). Source data are provided as a Source Data file.

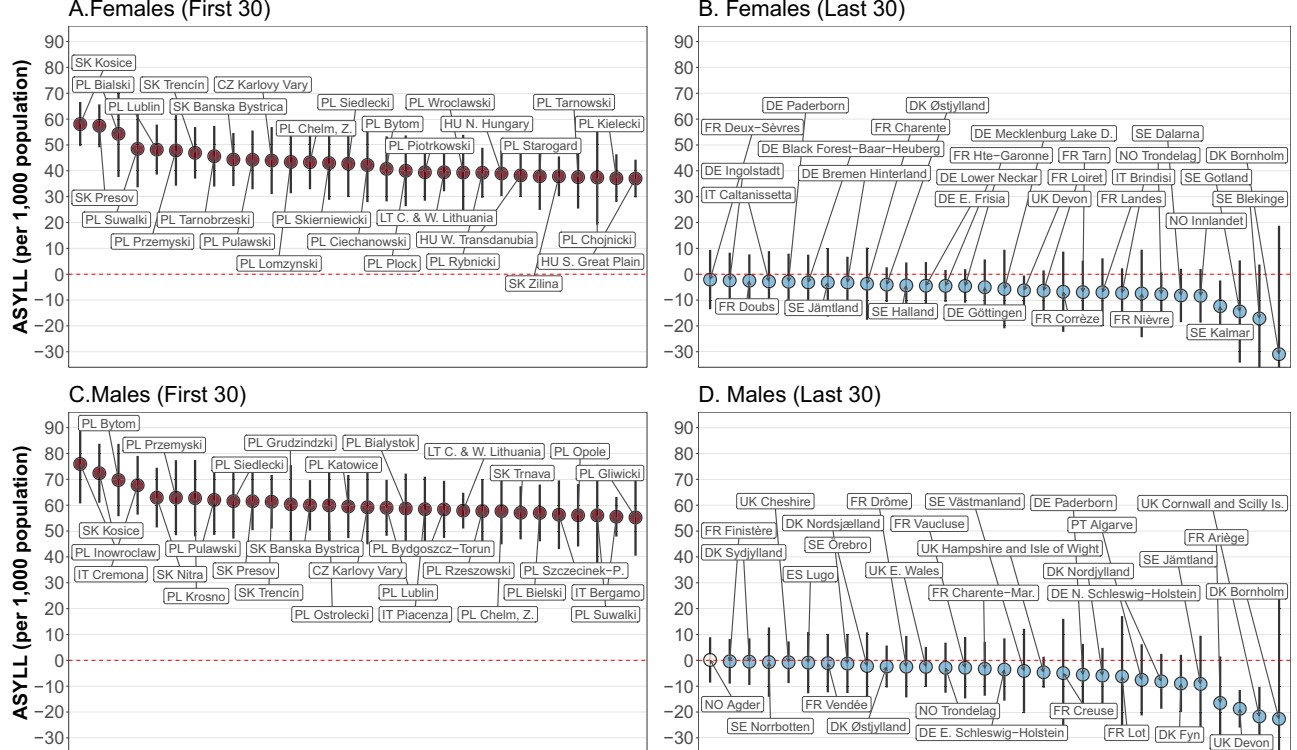

**Fig. 5 | Highest and lowest values of age-standardised years of life lost (ASYLL) in 2020–2021 (per 1000 population).** Notes: Vertical bars represent 95% confidence intervals. **A**, **B** show the first 30 regions of the ASYLL ranking with respect to the highest or lowest ASYLL for the female population. **C**, **D** Show the corresponding values for the male population. ASYLL quantifies the level of excess mortality in terms of life years lost (see the notes of Fig. 1 for more details). Source data are provided as a Source Data file.

The panels of Fig. 4 provide a clear illustration of the so-called Simpsons' paradox[34]. If the association is examined across all spatial units without stratifying them into CEE and West, it appears to be negative as highlighted by the fitted regression line in black. That is, the higher initial level of life expectancy observed in 2015–2019, the lower ASYLL is. In 2020, this association was rather weak, but it became strong in 2021. However, once East and West are analysed separately, it becomes apparent that there is no clear relationship between the two outcomes. In 2020, we can observe a positive association between initial life expectancy and ASYLL for both men and women in the West as well as women in the CEE (but not men). The results for 2021 are even more inconclusive. In the CEE stratum, there is a modest negative association among females, while there is hardly any among males. The same can be said about women in the West. Contrary to the other strata, we observe a positive association between the level of life expectancy at the baseline and the years of life lost in 2021 among men living in the West.

Figure 5 highlights the vanguard and laggard regions of Europe in terms of overall losses over the two pandemic years. The 'First 30' label refers to the spatial units having the highest losses, while the 'Last 30' label designates the European regions with no losses or gains in ASYLL during 2020–2021. In total, the 60 highlighted units constitute roughly 10 percent of the total number of the analysed regions.

The group of the most affected regions of Europe (panels A and C for women and men, respectively) is dominated by districts located in Eastern Europe, particularly in Poland and Slovakia. Among men, however, there are two Italian provinces with very high ASYLL values, Cremona (57.1 years per 1000 population, with CI ranging from 49 to 65.2) and Bergamo (51.7 years, CI 46.7 to 56.7). Unlike the laggard group of European regions, the vanguard group (panels B and D of Fig. 5) is quite heterogeneous. It consists of areas located in different parts of Europe (except CEE countries). Nevertheless, the majority of

best-performing regions belong to the Scandinavian countries, Germany, and France. In contrast to years of life losses, which were more pronounced among men, there are no notable differences between the sexes in years of life gains during the two pandemic years.

## Discussion

This study uses a large set of 569 small territorial units in 25 European countries to provide estimates of the total burden of the COVID-19 pandemic in both years, 2020 and 2021. Computing excess mortality at a fine geographical level, as opposed to a national level, is of paramount importance in understanding the pandemic's true impact. Fine-grained spatial analysis allows for a more accurate and nuanced assessment of the disparities in excess mortality, which are often masked when considering national aggregates.

Our findings using Age-Standardised Years of Life Lost (ASYLL) as the main indicator to measure excess mortality show evidence of gains in only seven regions in 2020, and four regions in 2021, for both sexes combined. On the contrary, 362 regions experienced notable losses during the first year of the pandemic, and 440 regions in the second year. Importantly, only eight regions suffered from ASYLL higher than 20 per 1000 population in 2020, while 75 regions suffered such a loss in 2021.

Our research additionally highlights significant regional variations in excess mortality within certain countries in both years. For instance, in Italy in 2020, our calculations did not show any excess mortality for both sexes combined in the Caltanissetta, Trapani and Potenza provinces, whereas our indicator reaches over 38 per 1000 population in Bergamo and Cremona. Similarly, in 2021 in Germany, our calculations showed an almost significant gain in East Schleswig-Holstein for males, while our indicator reached over 16.5 per 1000 population in South Saxony and North Thuringia. Finally, during the two years studied, in Poland, our study shows a maximum excess mortality of 32 per 1000

population in the Poznan region (West) for both sexes combined, while this value is at least 60 per 1000 population in the Pulawski region (East).

Beyond these country-specific case studies, the contiguity of the regions investigated in our study enables us to explore a vast part of Europe and to reveal the geographical patterns of the pandemic during 2020 and 2021, which are quite different. This approach is novel because most previous studies have focused primarily on single-country regions or on regions in various countries with no common borders. During the initial pandemic year, high values for excess mortality were concentrated in northern Italy, southern Switzerland, central Spain, and Poland, aligning with the early COVID-19 outbreak areas. Notably, most of the French and German regions as well as Finland, Iceland, Northern Ireland, southern Great Britain, Estonia, Latvia, and Hungary experienced comparatively lower losses. The spatial dynamics of excess mortality in Europe underwent a significant shift in the second pandemic year.

In 2021, Eastern European countries, particularly Slovakia, Hungary, and Latvia, showed the highest losses, following a discernible East-West mortality gradient. Using our regional values, we reveal that there is no relation between excess mortality in 2021 and pre-pandemic level of mortality when disentangling values between Western Europe and CEE countries This outcome is in line with the results of a recent study[33], who concluded that the East-West differences in excess mortality are related to structural and psychosocial traits that have their roots in the communist era. On the one hand, this includes differences in the connectivity of populations, driving the later onset of the pandemic in the East (from October 2020 onwards), while the West was hit more in the first wave (March to May 2020). On the other hand, this likely includes profound disparities in levels of vulnerability to the disadvantage of the East, e.g. in terms of pre-existing diseases, intensified by lagging economic development and selective migration due to their impact on risk-relevant behaviour. Lower levels of compliance with policy interventions (e.g. social distancing and vaccination) and a generally lower level of trust in authorities might also stem from the communist past of CEE countries.

The two pandemic years highly impacted male mortality in Eastern European countries (Poland, Slovakia, and Lithuania), which experienced high ASYLL values comparable to northern Italy and central Spain. Estonia exhibited the lowest male excess mortality among the Baltic States. Regions in Scandinavia, western and northern Germany, and southern France experienced relatively modest losses or gains, particularly among women.

In contrast to other investigations regarding COVID-19-related mortality, we calculated excess mortality for the 569 European regions of our panel utilising official mortality data regularly gathered by vital registration systems, which are less susceptible to reporting delays and misclassification. We ensured the consistency of the sum of regional data for all age groups within each country by cross-referencing it with information from the Human Mortality Database. This validation is particularly essential not only for computing outcomes in the context of older age groups, for which single-year-of-age data necessitates statistical techniques, but also due to the heightened vulnerability of older individuals to COVID-19.

Furthermore, we determined the baseline mortality using an up-to-date statistical approach that optimises the time frames in our models for projecting regional trends in 2020 in 2021. By aggregating our outcomes at national level, we conducted a comparative analysis with a previous study to validate the reliability of our findings: our estimations concerning declines in life expectancy generally align with the results of Schöley et al.[17]. In cases where disparities exist (Switzerland, Spain, Estonia, Lithuania), they are primarily associated with the year 2021 and the observed life expectancies (see Table A2 for specifics).

We use ASYLL to estimate the comprehensive burden of COVID-19 in 2020 and 2021 instead of common measures such as life expectancy and age-standardised death rates since they cannot be added up over time. ASYLL is a measure used in public health and epidemiology to assess the impact of premature mortality on a population: it measures the cumulative years of life lost attributed to a specific cause of death and, in this context, to a particular crisis. ASYLL facilitates meaningful comparisons across diverse populations by accommodating variations in age structures.

Our study focuses exclusively on the years 2020 and 2021 due to data availability. However, it is widely accepted that excess mortality in 2020 and 2021 were driven by the COVID-19 pandemic, which was not true for 2022 anymore. European countries and the global community have gained a better understanding of the virus, its variants, and effective measures to control its spread. Moreover, a substantial proportion of the population in European countries has been vaccinated by 2022, likely reducing the severity of illness and the case fatality related to COVID-19. Finally, there were severe influenza waves in late 2022 that probably contributed significantly to the regional levels and variation of mortality. Therefore, it would be inappropriate to attribute excess mortality in 2022 fully to the impact of COVID-19 and mix it with 2020 and 2021.

Although our chosen forecasting approach is robust and adaptable to various demographic scenarios and smaller populations, we model each geographical unit independently and do not account for spatial autocorrelation. Incorporating spatial structure into mortality modelling and forecasting may eventually reduce uncertainties surrounding excess mortality estimates. To our knowledge, only two previous studies have taken into account spatial dependence in this context, specifically modelling weekly mortality data and addressing challenges unique to that framework[29,35]. Dealing with yearly mortality data, we place greater emphasis on refining the time windows employed in forecasting expected mortality in the absence of a pandemic. Sensitivity analyses and comparisons of various alternative approaches for estimating baseline mortality levels have been proposed across diverse data structures[36–39]. Exploring these alternatives could provide additional insights. Moreover, our methodology cannot adjust for the potential harvesting effect following the first waves of the pandemic in each region, which could alter the mortality rates in 2021. To address this limitation and provide a more comprehensive understanding of the long-term impact of the pandemic on mortality, this analysis should be enlarged by incorporating causes of death. This constitutes a promising avenue for research in the future.

Finally, our research contributes to the prevailing body of literature concerning excess mortality in the context of the COVID-19 pandemic in both 2020 and 2021. We distinguish our study by presenting findings at the regional level for numerous European countries, a dimension that has hitherto not well been explored. The results underscore the significance of conducting a regional analysis, as we demonstrate that national-level estimates would obscure notable regional variations at least for 2020. It is imperative for policymakers to recognise this intranational heterogeneity to comprehensively evaluate the pandemic's impact within their respective countries and formulate health policy responses that are tailored to specific regional needs. Our results confirm that the pandemic affected urban areas that are particularly connected with international trade and travel (i.e. transit hubs) first, from where it spread to less connected peripheral areas and most of Eastern Europe, especially after policy interventions were loosened. From this, we can conclude that rapid interventions that limit the connectivity of important transit hubs, especially towards world regions that experienced an outbreak of a novel transmittable disease, appear most promising to prevent an epidemic from turning into a pandemic.

This research paves the way for two promising avenues. First, while our study effectively quantifies regional variations in excess

mortality, it does not offer insights into their underlying causes. This would involve associating these estimates with both regional contextual factors and public policies related to social distancing and international isolation, which were implemented at both regional and national levels. Then, ecological analyses could be conducted in parallel to well-designed epidemiological studies. This approach would enable the identification of key factors that account for the regional differences we have identified, leading to a deeper understanding of how to manage the transmission of a new infectious disease. For instance, the notably high mortality observed near Bergamo and Cremona in Italy could be attributed to the early onset of the pandemic, which prompted a robust public response from the Italian government which spared the southern regions of the country.

Second, comparing excess mortality due to COVID-19 with other historical mortality crises could be a valuable analytical tool for placing this pandemic in a broader historical and public health context. Such comparisons offer insights into the uniqueness and severity of the impact of COVID-19 by drawing parallels or distinctions with past crises such as influenza pandemics, major wars, or other epidemics. This comparative approach should help researchers and policymakers better understand the relative gravity of the pandemic, assess the efficacy of response measures, and identify patterns that might inform future preparedness efforts.

## Methods
### Data preparation
We collected subnational death and population counts for 25 European countries by age classes and sex from Eurostat, the Human Mortality Database[40] and national statistical offices. To ensure comparability of the selected spatial units in size and structure, we relied mostly on the Nomenclature of Territorial Units for Statistics (NUTS), using NUTS-3 levels for Czechia, Denmark, France, Italy, Luxembourg, Poland, Slovakia, Spain and Sweden, NUTS-2 for Austria, Belgium, Estonia, Finland, Hungary, Iceland, Latvia, Lithuania, the Netherlands, Norway, Portugal, Slovenia, Switzerland, England and Wales, as well as NUTS-1 for Ireland, Northern Ireland and Scotland. For Germany, we applied a national spatial classification ("Raumordnungsregionen")[41]. Minor adjustments had to be made due to territorial changes over time and data availability issues (see Supplementary Table 1 in online supplementary Appendix A for details). To verify the data quality, we compared our data obtained at regional level with data from the Human Mortality Database when available; differences are negligible.

Because of varying age classes in these data, we harmonised them into single-year age intervals up to 95+ for all spatial units[42]. The lowest number of age groups in our input data is eighteen (for Germany) and the largest age group that we ungrouped into single years of age is fourteen (for Germany, deaths at age 1 to 14).

In total, we analysed 569 harmonised spatial units containing populations ranging from 40,000 (Bornholm, Denmark) to 6,750,000 (Madrid, Spain).

### Methodology
When addressing the issue of excess mortality, a central methodological challenge involves estimating the baseline mortality level, which represents what would have been expected in the absence of the pandemic. Often, pre-pandemic mortality levels are used as the baseline due to their ease of acquisition and computation. However, this simplistic approach often overlooks temporal trends. To establish a more appropriate expected mortality level in the absence of COVID-19, it is necessary to use pre-pandemic historical trends for forecasting the pandemic-affected year, such as 2020 or 2021.

Among the various methodologies available (e.g[43].), we chose to use a *CP*-spline approach[44], combining two-dimensional *P*-splines with prior demographic insights derived from historical patterns specific to each population.

One significant advantage of employing a non-parametric approach like *CP*-splines is its remarkable flexibility in describing diverse mortality scenarios, which is especially valuable when dealing with 569 distinct subpopulations across 25 European countries. Additionally, it ensures the generation of smooth and plausible age profiles and time trends, while enhancing robustness when analysing smaller populations at risk.

Moreover, instead of utilising all available data uniformly for each region or exclusively relying on the common last available years, we fine-tune region-specific timeframes to forecast values for 2020 and 2021. In this way, we validate the mortality trends specific to each region, which we ultimately incorporate into our projections. In practice, leveraging the relatively low computational costs associated with CP-splines, we implement our method with a rolling starting year up to 2010. We then forecast 2019, measuring the distance between the observed and forecasted 2019 mortality. Working in a Poisson setting, we opt to measure distance by deviance[45]. The starting year with the lowest deviance value was selected for the final analysis. More information about this approach can be found in ref. 39.

It is worth noting that this entire procedure can be applied to any age group and is applicable regardless of the mortality indicator chosen for estimating excess mortality, such as life expectancy or age-standardised death rates. However, these indicators are not good candidates to estimate the total burden of the pandemic while differentiating 2020 from 2021, as they cannot be added up over time. Therefore, we chose to use Age-Standardised Years of Life Lost (ASYLL) to do so.

ASYLL is a measure used in public health and epidemiology[46] to assess the impact of premature mortality on a population: it quantifies the cumulative years of life lost attributable to a particular cause of death and can be adjusted to quantify the total years of life lost in the context of a specific crisis. Moreover, this metric allows for meaningful comparisons between different populations, as it accounts for variations in age structures. In few words, computing ASYLL involves (1) to identify the number of excess deaths within each age group, (2) to calculate for each deceased the number of years they would have been expected to live if they had not experienced premature death, (3) to sum the years of life lost for all individuals in each age group, and (4) to implement age-standardisation by adjusting the years of life lost in each age group considering a standard population's age distribution. This standard population is chosen to represent a hypothetical population with a fixed age structure, facilitating more meaningful comparisons between different populations; we used the 2013 European Standard Population (ESP). As an example, an ASYLL value of 20 indicates a standard population of 1000 inhabitants have experienced a loss of 20 years of life.

Additional details on the analytical procedure can be found online in supplementary appendix B. Supplementary appendix C contains a data visualisation tool and detailed values of our estimates for ASYLL and life expectancy at birth, along with confidence intervals. Moreover, we provide in Supplementary Appendix C both the data and code needed to replicate our estimates for the 95 French NUTS-3 regions, and these resources can be readily adapted for any available mortality data.

All calculations were carried out using R version 4.3.1[47]. In addition, we used ArcGIS 10.8.1[48] to merge the NUTS shapefile from Eurostat with the German 'Raumordnungsregionen' shapefile from the Federal Agency for Cartography and Geodesy (BKG), to apply the territorial adjustments stated in Table A1, and to construct the maps for this paper.

### Reporting summary
Further information on research design is available in the Nature Portfolio Reporting Summary linked to this article.

## Data availability

**Austria** Raw mortality data files at the level of Austrian Bezirke can be requested from Statistik Austria https://www.statistik.at/en/databases/statcube-statistical-database. Population data is available at Eurostat: https://ec.europa.eu/eurostat/web/main/data/database **Belgium** The selected aggregation by age, sex and NUTS2 regions is subject to a request at Statbel: https://statbel.fgov.be/en **Czechia** Access to mortality data is subject to a request at the Czech statistical office. Population data is available at: https://vdb.czso.cz/vdbvo2/faces/en/index.jsf?page=uziv-dotaz# **Denmark** Population and death counts are available at Statistics Denmark: https://www.statbank.dk **Estonia** Population and death counts are available at Statistics Estonia: https://andmed.stat.ee/en/stat **Finland** Population and death counts are available at Statistics Finland: https://pxdata.stat.fi/PxWeb/pxweb/en/StatFin/ **France** Population and death counts have been collected within the Human French Mortality Database project: https://frdata.org/en/french-human-mortality-database/ **Germany** Detailed death counts for German regions can be requested for a fee at the research data center of the statistical offices of the German Länder https://www.forschungsdatenzentrum.de/de/gesundheit/todesursachen. Population data can be requested at the federal statistical office https://www.destatis.de/EN/Service/Contact/_Contact.html. **Hungary** Raw mortality data files are available at the Hungarian Central Statistical Office: https://statinfo.ksh.hu/Statinfo/themeSelector.jsp. Population counts are available at Eurostat: https://ec.europa.eu/eurostat/web/main/data/database. **Iceland and Ireland** Death and population counts are available at the Human Mortality Database: https://www.mortality.org/. As data for 2021 was not yet available at HMD, we added data from Eurostat for this period: https://ec.europa.eu/eurostat/web/main/data/database. **Italy** Raw Data on death and population counts are available at Istat: https://www.istat.it/en/population-and-households?data-and-indicators **Latvia** Death and population counts are available at the Official Statistics: Portal Latvia: https://stat.gov.lv/en/statistics-themes/population **Lithuania** Population and death counts are available at Eurostat: https://ec.europa.eu/eurostat/web/main/data/database **Luxembourg** Death and population counts are available at the Human Mortality Database: https://www.mortality.org/. **Netherlands** Death and population counts are available at Statistics Netherlands: https://opendata.cbs.nl/statline/#/CBS/nl/. **Norway** Death and Population counts are available at Statistics Norway: https://www.ssb.no/en/statbank **Poland** Death and Population counts are available at Statistics Poland: https://bdl.stat.gov.pl/bdl/start **Portugal** Death counts are available at Statistics Portugal: https://www.ine.pt/xportal/xmain?xpid=INE&xpgid=ine_base_dados&contexto=bd&selTab=tab2&xlang=en. Population data is available at Eurostat: https://ec.europa.eu/eurostat/web/main/data/database. **Slovakia** Death and Population counts are available at the Slovakian Statistical Office: https://datacube.statistics.sk/. **Slovenia** Death and Population counts are available at the Slovenian Statistical Office: https://pxweb.stat.si/SiStat/en. **Spain** Death and Population counts are available at the Spanish Statistical Office: https://www.ine.es/en/. **Sweden** Death and Population counts are available at Statistics Sweden: https://www.scb.se/en/. **Switzerland** Population counts are available at the Federal Statistical Office: https://www.pxweb.bfs.admin.ch/pxweb/en/. Death counts are available at Eurostat: https://ec.europa.eu/eurostat/web/main/data/database. **United Kingdom** Raw population counts for England and Wales are available at the Office for National Statistics: https://www.ons.gov.uk/peoplepopulationandcommunity/populationandmigration/populationestimates/datasets/populationestimatesforukenglandandwalesscotlandandnorthernireland. Raw death counts used England and Wales are available at the Office for National Statistics: https://www.ons.gov.uk/peoplepopulationandcommunity/healthandsocialcare/causesofdeath/datasets/deathregistrationsandoccurrencesbylocalauthorityandhealthboard. Death and population counts for Northern Ireland and Scotland are available at the Human Mortality Database: https://www.mortality.org/. Our maps are based on shapefiles publicly available at Eurostat and the Federal Agency for Cartography and Geodesy (BKG). Detailed values of our estimates for the 569 regions are available at: https://osf.io/fwtsa/?view_only=ba00308358dc4fbaa23de72f9c82d1db. Source data used to produce our figures are provided with this paper. Source data are provided with this paper.

## Code availability

The.R code needed to replicate our estimates for the 95 French NUTS-3 regions is available at: https://osf.io/fwtsa/?view_only=ba00308358dc4fbaa23de72f9c82d1db.

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

## Acknowledgements

We are very grateful to Markéta Majerová, Rok Hrzic, Magdalena Muszyńska-Spielauer, and Mathias Lerch for providing, respectively, Czech, Slovenian, Austrian, and Swiss data. I.A., P.G., M.M., and M.S. were supported by funding from the European Research Council (ERC) under the European Union's Horizon 2020 research and innovation programme (grant agreement No 851485).

## Author contributions

Conceptualization: F.B., M.S., P.G., C.G.C., M.M. Methodology: C.G.C., F.B., P.G. Data collection and curation: M.S., F.B., I.A., M.M., P.G., C.G.C. Formal analysis: F.B., P.G. Writing - Original draft: P.G., F.B., C.G.C. Writing - Review & Editing: P.G., F.B., C.G.C., M.S., M.M., I.A. Visualisation: M.S., F.B., P.G.

## Competing interests

The authors declare no competing interests.
