## [Peer Review File · Nature Communications]

Spatial Disparities in the Mortality Burden of the Covid-19 pandemic across 569 European Regions (2020-2021)REVIEWER COMMENTS

Reviewer #1 (Remarks to the Author):

The authors have produced a clearly constructed report of their findings on an important topic. The results are noteworthy, particularly the east west variation seen across Europe over the two pandemic years. However, there are some areas that require some clarity or further explanation within the manuscript.

Major comments.

The methodological approach appears robust and is presented clearly- however there are some areas where additional information would be helpful.

In the methods the author's state:

"Often, pre-pandemic mortality levels are used as the baseline due to their ease of acquisition and computation. However, this simplistic approach often overlooks temporal trends. To establish a more appropriate expected mortality level in the absence of COVID-19, it is necessary to use pre-pandemic historical trends for forecasting the pandemic-affected year, such as 2020 or 2021"

Which years did you use to create your baseline estimates? Was this the same for all regions? If there were years of unusual mortality levels within the baseline period for one region/age group- how was this dealt with?

Are the sensitivity analyses around the calculation of the baseline available?

The authors repeat the statement that their work is the first to compare excess mortality at a regional level and compare using an appropriate metric. However, this report comes to mind Comparisons of all-cause mortality between European countries at local levels - Office for National Statistics. As does some of the work done by Euromomo- for whom I believe data is submitted at a regional level. It would be worth noting the work of these groups and how what you have done differs.

One fairly important limitation, but one not easy to address is the failure to adjust for mortality displacement following the first waves of mortality in each region. This should be acknowledged. Similarly – the relatively crude nature of ASYLL which assumes deaths were distributed equally within age/sex groups (years) – should also be mentioned. This could also be interpreted in relation to the variation between east and west

Minor comments

Abstract line 2- we produce age-sex specific

Abstract line 5 – it's not clear in this sentence that the forecasting method is used to estimate excess mortality and whether the losses stated related to ASYLL or excess deaths.

Introduction – paragraph 2, line 5. I assume this sentence is implying variation in COVID-19 testing between regions but this is not explicitly stated. At the end of this sentence the authors state: "diverse coverage by place of death". I am unsure what this means.

Results paragraph 7 – "The analysis of the combined effects of the two pandemic years indicates that excess mortality was generally higher in the countries of Central and Eastern Europe (CEE) having lower life expectancy."

Some clarity is needed here. Perhaps 'that have lower life expectancy'?

Figure 2 2020 appears to be identical to figure 1 2020

Discussion – given that you have regional data – was there a consideration of rural and urban? I appreciate this is leaning further in to the 'why'

Reviewer #2 (Remarks to the Author):

see attached pdf

NCOMMS-23-62194

Beyond Borders: Spatial Disparities in the Mortality Burden of the Covid-19 pandemic across 569 European Regions (2020-2021)

Summary and comments to the authors

The above-mentioned manuscript examines the effect of the COVID-19 pandemic at the sub-regional level in Europe. The authors have used vital mortality registration data during 2020 and 2021 and estimated Age-Standardised Years of Life Loss (ASYLL). They reported large spatial variation at the subregional level with a clear East-West gradient in 2021. I think the paper interesting and a nice summary of the impact of the pandemic at the sub-regional level in Europe. I have the following comments:

- Methodological considerations
 - As mentioned by the authors, mortality is estimated independently per region. The authors mention that there are no methods for forecasting regional mortality while accounting for spatial dependence. This is wrong. Previous studies examining excess mortality in the subregional level have accounted for spatial dependence (mainly in the Bayesian framework, see Blangiardo et al 2020 PlosONE and Konstantinoudis et al 2022 Nature Communications). One can also incorporate higher order of spatial interactions to account potential different trends per region. This needs to be acknowledged and authors need to provide a justification for their choice not to account for it.
 - The authors mention the use of two-dimensional P-splines to model mortality. The paper should stand itself, and thus they should provide a more detailed explanation of the method in the supplement. The flexibility of the splines is an attractive property, but at the same time, could lead to overfit, as the case of Germany in the WHO estimates. This also needs to be discussed in the paper.
 - The authors need to validate the estimates. I suggest an internal cross validation; leave out the past two years for which you have data, and calculate coverage probability, bias and mean square error by area. I also suggest an external validation; calculate the national estimates of expected or excess mortality and compare them with published studies by year.
- Data and code should be provided in github.

- You need to provide some uncertainty estimates by subregion.
- I do not understand what the association between ASYLL and age expectancy shows, given that the ASYLL is age adjusted. Can you formulate your hypothesis behind testing this association?
- Although the authors provide justification of not including 2022, as the end of the pandemic was May 2023 it would be nice to provide estimate for the entire period of the pandemic, acknowledging potential differences in the interpretation.
- The authors should elaborate on the importance and the public health message of this paper, given on the large number of publications already assess the impact of COVID-10 mortality. What does this mean for stakeholders in the future? How could we focus on mostly impacted societies and what should we do?

Response to Reviewer 1

Dear Referee,

Thank you so much for your comments and suggestions, which helped us to substantially improve our manuscript.

- 1. In the methods the author's state: "Often, pre-pandemic mortality levels are used as the baseline due to their ease of acquisition and computation. However, this simplistic approach often overlooks temporal trends. To establish a more appropriate expected mortality level in the absence of COVID-19, it is necessary to use pre-pandemic historical trends for forecasting the pandemic-affected year, such as 2020 or 2021". Which years did you use to create your baseline estimates? Was this the same for all regions? If there were years of unusual mortality levels within the baseline period for one region/age group- how was this dealt with? Are the sensitivity analyses around the calculation of the baseline available?***

It is true that in our initial submission we were not perfectly clear about our methodological approach. To address this concern, now we provide a comprehensive description of the entire methodology in the online Supplementary Appendix B. Furthermore, we have added the following paragraph to the main text (p.16) to clarify this point:

"Moreover, instead of utilizing all available data uniformly for each region or exclusively relying on the common last available years, we fine-tune region-specific timeframes to forecast values for 2020 and 2021. In this way, we validate the mortality trends specific to each region, which we ultimately incorporate into our projections. In practice, leveraging the relatively low computational costs associated with CP-splines, we implement our method with a rolling starting year up to 2010. We then forecast 2019, measuring the distance between the observed and forecasted 2019 mortality. Working in a Poisson setting, we opt to measure distance by deviance [45]. The starting year with the lowest deviance value was selected for the final analysis. More information about this approach can be found in [39]."

An appealing feature of our methodological approach is that it does not assume any arbitrary choice concerning the length of the historical mortality trends. Therefore, conducting a classical sensitivity analysis under different scenarios (i.e., 5 years, 10 years of historical trends) within this framework is not feasible a priori. Nevertheless, we are convinced that the sensitivity analysis is indeed inherently carried out within the method routine through optimizing the time window used for forecasting. We calculate potential outcomes for all possible numbers of time windows in each region, formulate an objective function for minimization, and showcase only the optimal time window.

- 2. The authors repeat the statement that their work is the first to compare excess mortality at a regional level and compare using an appropriate metric. However, this report comes to mind Comparisons of all-cause mortality between European countries at local levels - Office for National Statistics. As does some of the work done by Euromomo- for whom I believe data is submitted at a regional level. It would be worth noting the work of these groups and how what you have done differs.***

Many thanks for bringing our attention to the work from ONS, which is now cited in the introduction. More specifically, we have added the following text (p. 2):

“Another peer-reviewed study covers 200 NUTS 2 European regions for the years 2020 to 2022 but does not estimate excess mortality in regions of Germany, the UK, Ireland or Sweden [30]. The ONS has also published on its website a report [31] on weekly excess mortality between the end of 2019 and mid-2022 by region in Europe.”

- 3. One fairly important limitation, but one not easy to address is the failure to adjust for mortality displacement following the first waves of mortality in each region. This should be acknowledged.**

The issue of mortality displacement/harvesting effects is indeed a valid point. In fact, this was the main motivation for us to stick to 2020 and 2021 only. We agree that the issue of mortality displacement should be acknowledged. Consequently, we have added the following paragraph in the discussion (p.14)

“Moreover, our methodology cannot adjust for the potential harvesting effect following the first waves of the pandemic in each region, which could alter the mortality rates in 2021. To address this limitation and provide a more comprehensive understanding of the long-term impact of the pandemic on mortality, this analysis should be enlarged by incorporating causes of death. This constitutes a promising avenue for research in the future.”

- 4. Similarly – the relatively crude nature of ASYLL which assumes deaths were distributed equally within age/sex groups (years) – should also be mentioned. This could also be interpreted in relation to the variation between east and west**

We are not sure if we understand this comment correctly. Our measure of ASYLL (age-standardized years of life lost) is not crude by definition, as it adjusts for variation in age structures between different population sub-groups. We calculate ASYLL for each year, region, and sex through the direct method standardization, and using the 2013 European Standard Population as a standard age distribution (more details are provided in ‘Data and Methods’ section). If this comment refers to the inability of ASYLL to disentangle harvesting effects, then it is also applicable to any other summary measure of mortality. The certain advantage of using ASYLL over alternative mortality measures is its feature to be additive. This mitigates somehow the issue of mortality displacement through estimating the cumulative impact of the pandemic over 2020 and 2021.

5. Minor comments

5.1. Abstract line 2- we produce age-sex specific

5.2. Introduction – paragraph 2, line 5. I assume this sentence is implying variation in COVID-19 testing between regions but this is not explicitly stated. At the end of this sentence the authors state: “diverse coverage by place of death”. I am unsure what this means.

5.3. Results paragraph 7 – “The analysis of the combined effects of the two pandemic years indicates that excess mortality was generally higher in the countries of Central and Eastern Europe (CEE) having lower life expectancy.” Some clarity is needed here. Perhaps ‘that have lower life expectancy’?

5.4. Abstract line 5 – it's not clear in this sentence that the forecasting method is used to estimate excess mortality and whether the losses stated related to ASYLL or excess deaths.

We have changed the sentences accordingly.

5.5. Figure 2 2020 appears to be identical to figure 1 2020

We provide the figures in the resubmission of our paper.

5.6. Discussion – given that you have regional data – was there a consideration of rural and urban? I appreciate this is leaning further in to the 'why'

Following comment 8 of reviewer 2, we have changed one paragraph in the discussion, which introduces your point about rural/urban regions.

Response to Reviewer 2

Dear Referee,

Thank you so much for your comments and suggestions, which helped us to substantially improve our manuscript.

- 1. As mentioned by the authors, mortality is estimated independently per region. The authors mention that there are no methods for forecasting regional mortality while accounting for spatial dependence. This is wrong. Previous studies examining excess mortality in the subregional level have accounted for spatial dependence (mainly in the Bayesian framework, see Blangiardo et al 2020 PlosONE and Konstantinoudis et al 2022 Nature Communications). One can also incorporate higher order of spatial interactions to account potential different trends per region. This needs to be acknowledged and authors need to provide a justification for their choice not to account for it.***

Thank you for your insightful comment and for bringing to our attention the studies by Blangiardo et al. (2020) and Konstantinoudis et al. (2022), which address the issue of spatial dependence in forecasting regional mortality. Considering your comment, we modified the text accordingly to acknowledge these existing studies and provide a clear justification for our choice not to explicitly account for spatial dependence in our specific context.

Specifically, it is important to highlight here that Blangiardo et al. concentrate solely on the age-adjusted expected number of deaths, while Konstantinoudis et al. deal with adult ages, treating age-groups as factors. Moreover, both studies employ weekly data modelling, introducing distinct challenges stemming from smaller sample sizes and seasonal patterns. This underscores the necessity of incorporating additional structure through the consideration of spatial dependence.

The aim of our work was to tackle the distinct challenges posed by yearly data encompassing all individual ages for the computation of ASYLL. Additionally, as elucidated by Bonnet and Camarda (2024), who provided the foundation for the methodologies implemented in our study, when dealing with regional units, a significant portion of uncertainty originates from the randomness inherent in observed data. As a result, incorporating spatial correlation into modelled and forecasted values is not expected to significantly alter point estimates and is unlikely to have a substantial impact on the significance of the conclusions.

The revised paragraph includes, among other elements, your input, and it is formulated as follows (p. 14):

“Although our chosen forecasting approach is robust and adaptable to various demographic scenarios and smaller populations, we model each geographical unit independently and do not account for spatial autocorrelation. Incorporating spatial structure into mortality modelling and forecasting may eventually reduce uncertainties surrounding excess mortality estimates. To our knowledge, only two previous studies have taken into account spatial dependence in this context, specifically modelling weekly mortality data and addressing challenges unique to that framework [29, 35]. Dealing with yearly mortality data, we place greater emphasis on refining the time windows employed in forecasting expected mortality in the absence of a pandemic.”

- 2. The authors mention the use of two-dimensional P-splines to model mortality. The paper should stand itself, and thus they should provide a more detailed explanation of the method in the supplement. The flexibility of the splines is an attractive property, but at the same time, could lead to overfit, as the case of Germany in the WHO estimates. This also needs to be discussed in the paper.***

In the light of this comment as well as the similar comments of Reviewer 1 and the Senior Editor, we provide now a comprehensive description of the entire methodology in the online Supplementary Appendix B.

Regarding the potential risks associated with using splines in our context, we acknowledge the German study conducted by WHO researchers. In March 2022, scholars working in collaboration with the WHO released an update on SARS-CoV-2-related excess mortality in Germany for the years 2020 and 2021, reporting a total of 194,988 over the two-year period. Detailed information on their methodology is available in both Msemburi et al. (2023, Nature 613, 130-137) and Knutson et al. (2023, The Annals of Applied Statistics, 17, 1353-1374).

Subsequently, the authors, incorporating feedback from the WHO-UN Technical Advisory Group on COVID-19 Mortality Assessment¹, revised the total excess deaths to 122,000 over the same two-year span. It is noteworthy that, at the time of the initial publication, the German excess mortality was deemed particularly unusual, and the flexibility of the splines was identified as the potential cause for the overestimation. However, we emphasize that these studies calculated their figures using a Bayesian sampling model with spline-based seasonal variation, directly modelling excess mortality for all ages and incorporating a long set of covariates to predict estimated values for countries without data, aiming to provide estimates for all countries in the world.

In contrast, our approach involves mortality data for each region over single age and year, fully modelled without the use of external covariates. B-splines function solely as a regression basis for constructing the entire regional mortality surface. Subsequently, the imposition of a smoothness penalty guarantees a smooth outcome, leading to P-splines. Furthermore, incorporating asymmetric penalties on relative derivatives concerning age and time, grounded in observed trends, enforces constraints on future mortality, giving rise to what is referred to as CP-splines.

- 3. The authors need to validate the estimates. I suggest an internal cross validation; leave out the past two years for which you have data, and calculate coverage probability, bias and mean square error by area. I also suggest an external validation; calculate the national estimates of expected or excess mortality and compare them with published studies by year.***

Thank you for this remark. Regarding interval cross-validation, we are fortunate to have a long though highly diverse range of available years for each region. The spectrum spans from Sweden and France, where regional mortality data have been available since 1969 and 1970, to Latvia and Poland, where data is only accessible from 2002 and 2006. Including all available data for each region or, conversely, relying exclusively on the common last available years, a

¹ One of the authors of our paper was part of this group.

common practice in numerous previous analyses within this context, may lead to biased outcomes. Our objective is to steer clear of such biases.

As highlighted in our response to Comment 1 from Reviewer 1, we have thus chosen to optimize the time window for each region by determining the optimal number of observed years that would enhance the forecast for 2019, last pre-pandemic year. We are confident that this approach, thoroughly tested with French departments by Bonnet and Camarda (PLOS One, 2024), has the potential to mitigate potential biases stemming from the subjective selection of the time window for projecting mortality trends. Moreover, it explicitly incorporates interval cross-validation by fine-tuning the regional-based mortality trends we aim to consider in our projection. Additionally, we have chosen to emphasize this concept in the paper (p.16):

“Moreover, instead of utilizing all available data uniformly for each region or exclusively relying on the common last available years, we fine-tune region-specific timeframes to forecast values for 2020 and 2021. In this way, we validate the mortality trends specific to each region, which we ultimately incorporate into our projections. In practice, leveraging the relatively low computational costs associated with CP-splines, we implement our method with a rolling starting year up to 2010. We then forecast 2019, measuring the distance between the observed and forecasted 2019 mortality. Working in a Poisson setting, we opt to measure distance by deviance [45]. The starting year with the lowest deviance value was selected for the final analysis. More information about this approach can be found in [39].”

Furthermore, it is right that external validation of our results is essential. Accordingly, we have provided external validation (Table A2) by comparing our estimates of life expectancy losses at the national level with estimates coming from Schöley et al. (2022). Our findings show remarkably similar values, and we further discuss this comparison in the main text (p.13):

“By aggregating our outcomes at national level, we conducted a comparative analysis with a previous study to validate the reliability of our findings: our estimations concerning declines in life expectancy generally align with the results of Schöley et al. [17]. In cases where disparities exist (Switzerland, Spain, Estonia, Lithuania), they are primarily associated with the year 2021 and the observed life expectancies (see table A2 for specifics)”

4. Data and code should be provided in github.

Due to data confidentiality issues applied to some countries, we are not in the position to provide the complete dataset used in our analyses, and thus to ensure full reproducibility. However, we recognize the importance of being transparent. As an alternative solution, we provide now in the online supplementary appendix C the code and the input data for France, which enables users to follow the methodology and reproduce the output for this country, and any other country of their choice after getting access to the respective data. The sources and the format of the raw data used in our analyses are described in detail in Table A1.

5. You need to provide some uncertainty estimates by subregion.

We agree that confidence intervals are indeed essential when calculating excess mortality for European regions. Figure 5 provides confidence intervals around our regional estimates. Moreover, we have added estimated values along with confidence intervals in our online

supplementary appendix C. For both comments 4 and 5, we have added two sentences in the methodology section (p. 17):

“Additional details on the analytical procedure can be found online in supplementary appendix B. All calculations were carried out using R version 4.3.1 [47]. Supplementary appendix C contains a data visualisation tool and detailed values of our estimates for ASYLL and life expectancy at birth along with confidence intervals. Moreover, we provide in Supplementary appendix C both the data and code needed to replicate our estimates for the 95 French NUTS-3 regions, and these resources can be readily adapted for any available mortality data.”

6. I do not understand what the association between ASYLL and age expectancy shows, given that the ASYLL is age adjusted. Can you formulate your hypothesis behind testing this association?

We should have explained better the rationale behind examining the ecological associations between the magnitude of excess mortality. Our hypothesis is that regions with higher initial mortality during the pre-pandemic years (2015-2019) experienced higher excess mortality in 2020 and 2021. In this context, baseline life expectancy serves as a proxy of ‘initial conditions’ reflecting different aspects related to population health such as the quality and accessibility of health care, the level of socioeconomic development, the prevalence of risk factors and disease burden, environmental conditions, etc. We have added a respective clarification concerning this point to the main text (p.5):

“Our hypothesis is that regions with higher initial mortality during the pre-pandemic years (2015-2019) experienced higher excess mortality in 2020 and 2021. In this context, baseline life expectancy serves as a proxy of ‘initial conditions’ reflecting different aspects related to population health such as the quality and accessibility of health care, the level of socioeconomic development, the prevalence of risk factors and disease burden, environmental conditions, etc.”

7. Although the authors provide justification of not including 2022, as the end of the pandemic was May 2023 it would be nice to provide estimate for the entire period of the pandemic, acknowledging potential differences in the interpretation.

We would be happy to add 2022 to our analysis, even though some countries for which the regional mortality data are not yet available would not be covered. The major issue with adding 2022 is not the data availability but the comparability of excess mortality estimates with 2020-2021, and their interpretation. We firmly believe that it is hardly feasible to estimate the **direct** impact of the COVID-19 pandemic in 2022, at least using our methodological approach. Unlike 2020 and 2021, the year 2022 cannot be regarded as a solely pandemic year because of the impact of other factors such as the resumed influenza wave in Europe in the late autumn of 2022 (<https://www.ecdc.europa.eu/en>).

8. The authors should elaborate on the importance and the public health message of this paper, given on the large number of publications already assess the impact of COVID-10 mortality. What does this mean for stakeholders in the future? How could we focus on mostly impacted societies and what should we do?

Thank you for this comment. The last three paragraphs of the discussion already touched the field of policy relevance, but we agree that we were rather vague in this regard. Therefore, we have expanded the third paragraph from the last in the discussion. Now, it reads as follows:

“Finally, our research contributes to the prevailing body of literature concerning excess mortality in the context of the COVID-19 pandemic in both 2020 and 2021. We distinguish our study by presenting findings at the regional level for numerous European countries, a dimension that has hitherto not well been explored. The results underscore the significance of conducting a regional analysis, as we demonstrate that national-level estimates would obscure notable regional variations at least for 2020. It is imperative for policymakers to recognize this intranational heterogeneity to comprehensively evaluate the pandemic's impact within their respective countries and formulate health policy responses that are tailored to specific regional needs. Our results confirm that the pandemic affected urban areas that are particularly connected with international trade and travel (i.e. transit hubs) first, from where it spread to less connected peripheral areas and most of Eastern Europe, especially after policy interventions were loosened [28]. From this, we can conclude that rapid interventions that limit the connectivity of important transit hubs, especially towards world regions that experienced an outbreak of a novel transmittable disease, appear most promising to prevent an epidemic from turning into a pandemic.”

REVIEWERS' COMMENTS

Reviewer #1 (Remarks to the Author):

Thank you for addressing the comments so comprehensively.

With regard to the comment (4) about ASYLL - I was referring to the differences between the approach you have used and that of modelling (age sex and) comprehensive morbidity data as some COVID-19 population studies have done - few have extended this to estimate YLL (though it is possible). I acknowledge that this is not common in analysis of mortality trends, especially across regions as you have done so I am content with your justification.

Reviewer #2 (Remarks to the Author):

The authors have addressed all my concerns.